# Genetic Diversity and Molecular Characterization of Worldwide Prairie Grass (*Bromus catharticus* Vahl) Accessions Using SRAP Markers

Limei Yi [1] , Zhixiao Dong [1], Yu Lei [1], Junming Zhao [1], Yanli Xiong [1], Jian Yang [1], Yi Xiong [1] , Wenlong Gou [2] and Xiao Ma [1,*]

1   College of Grassland Science and Technology, Sichuan Agricultural University, Chengdu 611130, China; yilimei1997@126.com (L.Y.); dongzhixiao94@126.com (Z.D.); leiyu15908112551@163.com (Y.L.); junmingzhao163@163.com (J.Z.); yanlimaster@126.com (Y.X.); iwanwin@126.com (J.Y.); xiongyi95@126.com (Y.X.)
2   Sichuan Academy of Grassland Sciences, Chengdu 611743, China; gsz080115@163.com
*   Correspondence: maroar@126.com

**Abstract:** Prairie grass (*Bromus catharticus* Vahl) is an important grass species that could be used in the production systems of certified seed and high-quality forage for grazing ruminants. In the present research, a sequence-related amplified polymorphism (SRAP) marker was employed to detect the genetic variability and structure of 80 prairie grass accessions from all over the world. Altogether, 460 reliable bands were amplified from 47 SRAP primer pairs with 345 (75%) polymorphic bands. The average values of discrimination power (DP) and polymorphic information content (PIC) were 0.753 and 0.317, respectively. Both the UPGMA clustering and PCoA analyses grouped the 80 accessions into five clusters, whereas the STRUCTURE analysis showed that 80 prairie grass accessions possessed three genetic memberships (K = 3). The results of the Mantel test showed that the distance matrix has a moderately positive correlation between the morphological and molecular data sets (r = 0.524). A poor genetic differentiation (Fst = 0.045) was discovered among the six geo-groups of accessions. Besides, the highest intragroup genetic diversity was found in the North America group (He = 0.335). This study provides a genetic structure and diversity case for prairie grass, and supplies new clues for the study and utilization of prairie grass.

**Keywords:** SRAP; prairie grass; genetic diversity; germplasms

## 1. Introduction

The genus *Bromus* belongs to the tribe Bromeae which is part of subfamily Pooideae of the family Poaceae and consists of approximately 150–160 annual and perennial species of grasses that are distributed in temperate regions of the globe [1]. This genus shows a high incidence of polyploidy levels, ranging from diploid such as *B. tectorum* (2n = 2x = 14) to duodecaploid species such as *B. arizonicus* (2n = 12x = 84) [2]. One of the major agricultural species of forage grass is *Bromus catharticus* Vahl (synm. *Bromus unioloides* H.B.K. or *Bromus wildenowii* Kunth), also known as prairie grass or rescue grass [3]. *B. catharticus* is an annual or short-lived cold-season grass native to South America and has been widely introduced to and naturalized in Europe, Africa, Asia, Australia, and North America [4,5]. Prairie grass is classified in the section *Ceratochloa* of genus *Bromus* and is a hexaploid species (2n = 6x = 42) with the genomic formula AABBCC [1]. This species is recognized as an important grass resource not only in producing high-quality forage for the production systems of grazing ruminants but also in the production of certified seed. In addition, *B. catharticus* is potentially useful in soil and water conservation and cultivated pasture improvement due to its strong regeneration and adaptability to diverse environmental conditions.

Assessment of genetic relationships and diversity within the germplasm collection is of great importance for effective breeding and germplasm conservation. The genetic variation of prairie grass has been measured and characterized by various markers including morphological and agronomic traits and DNA molecular markers. Previous research on prairie grass pointed out substantial morphological variations between accessions or populations, plastic association with the climatic conditions of differing habitats and low heritability of vegetative and reproductive traits [5–9]. DNA molecular markers are preferred over traditional phenotypic and biochemical markers because they offer higher polymorphism with more precision and exhibit independence from the environment and the plant developmental phase [10]. Several molecular markers like random amplified polymorphic DNA (RAPD), amplified fragment length polymorphism (AFLP) and simple sequence repeats (SSR) have been used in elucidating the genetic diversity of prairie grass accessions [10–13]. Sequence-related amplified polymorphism (SRAP) is a PCR based dominant marker technique and is superior to common dominant markers such as RAPD, ISSR and AFLP markers, because of its simplicity, robustness, low cost and versatility. The principle behind its use is to specifically amplify coding regions of the genome with ambiguous primers targeting GC-rich exons (forward primers) and AT-rich promoters, introns, and spacers (reverse primers) [14,15]. Despite the lack of heterozygosity descriptors caused by data scored as dominant, SRAP has been successfully used to investigate patterns of genetic variability in plenty of grass species, including *Elymus breviaristatus* [16], *Stenotaphrum secundatum* [17], *Buchloe dactyloides* [18], *Dactylis glomerata* [19] and *Cynodon dactylon* [20].

*Bromus catharticus* is generally regarded as a predominantly self-pollinated(autogamous) species, with a natural outcrossing rate of only 1.8% [7]. The reproductive (mating) system could dramatically affect the biology and genetics of plant populations. Generally speaking, compared to the populations of outcrossing species, plants with a high self-crossing rate have higher levels of genetic differentiation among populations and lower levels of genetic diversity within populations [21,22]. Thus, understanding the genetic variability and structure within a broad germplasm collection could be significant for conservation of valuable genetic resources, selection of elite germplasms, and development of new cultivars. Although about 30 commercial cultivars were registered in OECD countries and China, the commercial importance of this species, and the real global patterns of genetic variability of *B. cartharticus* remain unclear due to limited studies and the use of a restricted number of accessions [10–12]. Such cases pose practical challenges in an increasing demand for new prairie grass cultivars with highly productive performance as well as strong adaptation to nonlocal environments and climate change.

This study described genetic variability and structure in 80 accessions of diverse prairie grass from a worldwide collection using SRAP markers. The objectives of this research were to: (1) reveal genetic differences and population structural characteristics of the accessions in the molecular level, and (2) obtain more comprehensive information on the genetic variation combining the morphological diversity detected in the previous study [9].

## 2. Materials and Methods

### 2.1. Plant Samples and DNA Extraction

Altogether, 80 prairie grass accessions were analyzed in the present study, including 8 collected from southwest China and 72 provided by the National Plant Germplasm System of USDA (NPGS) of the United States. There are four improvement status of the 80 germplasm, of which 33 were uncertain improvement (U), 4 were variety (V), 36 were wild (W) and 7 were cultivar (C) (Table S1). Phenotype evaluation has been described in our previous report [9]. Nine morpho-logical traits at the anthesis stage were scored in May 2015, in the experimental field in Ya'an, Sichuan Province, China. These nine traits were: plant height (PH, cm), length of first internode (LFI, cm), length of flag leaf (LFL, cm), length of second upper leaf (LSUL, cm), stem diameter (SD, mm), width of flag leaf

(WFL, cm), width of second up per leaf (WSUL, cm), dry matter yield (DMY, g plant-1) and tiller number (TN). Among them, PH, SD, LFI, LFL, LSUL, WFL, WSUL were measured 3 times per plant. The DMY and TN per plant can only be measured once. Phenotypic data are shown in the Table S2.

According to geographic origins, the germplasm collection could be subdivided into 6 priori defined groups: South America (22), North America (14), Africa (13), Europe (8), Oceania (8) and Asia (15) (Figure S1). All accessions germinated in a greenhouse (25/15 °C day/night temperature). Total genomic DNA bulked samples were made up from DNA extraction using equal amounts of fresh leaves from randomly selected 10 seedlings for each accession with a DNA extraction kit (Tiangen, Beijing, China). The density and quality of DNA were examined with a Nanodrop spectrophotometer (NanoDrop 2000C; ThermoScientific, Shanghai, China). Then, the extracted DNA were diluted into 20 ng/μL for SRAP analysis.

### 2.2. SRAP Analysis

A total of 400 SRAP primer combinations (20 forward and 20 reverse primers) were pre-screened using DNA samples of 8 accessions with a wide geographic origin and obvious morphological differences. Out of the 400 primer pairs, 47 primer pairs that generated clear polymorphic and reproducible bands were selected to evaluate polymorphism in 80 prairie grass accessions (Table S3). The SRAP-PCR amplification reactions were performed in a volume of 15 μL involving 2 μL (20 ng/μL) DNA samples, 0.6 μL (10 mM) forward and reverse primers, 7.0 μL 2× Master Mix (Tiangen, Beijing), 0.6 μL Taq enzyme (2.5 U/μL) and 4.2 μL ddH$_2$O. The cycling conditions were as follows: initiation step at 94 °C for 4 min, followed by 35× cycles of 1 min at 94 °C, annealing for 1 min at 52 °C, 2 min at 72 °C, and a final extension of 10 min at 72 °C. All PCR products were subjected to electrophoresis on an 8% non-denaturing polyacrylamide gel with 1× TBE buffer solution. Following gel electrophoresis, silver nitrate staining was employed to aid in visualizing polymorphic bands.

### 2.3. Data Analysis

Only intense, well-resolved, unambiguous bands (>50 bp) were manually scored as present (1) or absent (0) for creating binary matrix data. The number of polymorphic bands (NPB) was calculated and the power to identify differences of each SRAP marker was estimated by calculating the percentage of polymorphic bands (PPB), polymorphic information content (PIC), discriminating power (DP), marker index (MI), Nei's genetic diversity (H) and resolving power (RP).

PIC for dominant markers was estimated by applying the following formula:

$$\text{PIC} = 2\,\text{fi} \times (1 - \text{fi}) \tag{1}$$

where fi is the frequency with marker bands, and (1 − fi) is the frequency without marker bands [23]. The MI that was used to reflect the polymorphism information of each pair of primers was estimated as follows [24]:

$$\text{MI} = \text{PIC} \times \text{NPB} \tag{2}$$

The band informativeness (Ib) was measured using the following formula:

$$\text{Ib} = 1 - (2 \times |\,0.5 - \text{pi}\,|) \tag{3}$$

where pi is the amplification band frequency of the tested accessions [25]. RP was calculated as:

$$\text{RP} = \Sigma\text{Ib} \tag{4}$$

The discriminating power (DP) refers to the chance that two randomly selected individuals are characterized by different banding patterns and are thus distinguishable from

each other, and the DP value was calculated using the iMEC, an online marker efficiency calculator [26]. The GenAlex 6.51 program [27] was used to compute the allele number (Na), effective number of alleles (Ne), Shannon information index (I) expected heterozygosity (He), unbiased expected heterozygosity (uHe) and pairwise population PhiPT values (Fst) among the geographical groups.

At the germplasm level, the genetic similarity coefficient (Dice) was evaluated, and the Unweighted Pair Group Method with Arithmetic Mean (UPGMA) was conducted using NTSYS-pc software, then, principal coordinates analysis (PCoA) was performed. The relationship between morphology indices and genetic similarity coefficient of all the germplasm was measured using Mantel Test [28] and Procrustes analysis was performed in R (v 3.5.0) with vegan package [29]. A dendrogram was plotted by the Neighbor-Net Network method, performed with SplitsTree4 software after a 1000 bootstrap test using Jaccard Dice coefficient [30].

In addition, STRUCTURE software with a Bayesian model was applied to reveal the population structure [31]. The presumed number of genetic clusters (K value) was set from 1 to 10. This parameter includes a "Burnin Period" of 50,000 and "After Burnin" Markov Chain Monte Carlo replicates of 100,000. Three independent results were evaluated, and the number of the K value was acquired using the DeltaK method on the SRTUCTURE Harvester v.0.6.93 program. The optimal value of K was determined by CLUMPP1.1 software [32] and the results were illustrated by the GraphPad Prism 8.

## 3. Results

### 3.1. SRAP Genetic Diversity and Polymorphism in the Germplasm Collection

Genetic diversity among all assayed 80 accessions of prairie grass were assessed through 47 different SRAP primer combinations. Altogether, 460 reliable bands were amplified from these SRAP primers (Figure S2), with the number of reliable bands per primer set varying from 3 (Me8 + Em3) to 15 (Me10 + Em9, Me15 + Em6) (Table 1). The scorable band size ranged from 100 to 1100 base pairs. Of 460 scorable bands, 345 were identified as polymorphic (PPB = 75%). The number of polymorphic bands per primer set ranged from 2 (Me8 + Em16, Me20 + Em20) to 13 (Me9 + Em20), with an average of 7.3 bands. The percentage of polymorphic bands generated by each primer set was within the range of 33.3% (Me20 + Em20) to 100% (Me4 + Em8, Me4 + Em20, Me7 + Em7 and Me8 + Em3), with a mean of 75%. Furthermore, the indicators including PIC, MI, H, DP and RP were used to assess the polymorphisms and discrimination ability of the primers (Table 2). The mean value of PIC was 0.317, with the highest value of 0.425 in primers of Me2/Em3. The Shannon index (H) ranged from 0.190 to 0.5, with a mean of 0.465. The average values of D, MI and RP were 0.753, 2.314 and 3.387, respectively.

The Dice's similarity coefficients among the assayed accessions ranged from 0.5609 to 0.9804, with the average value of 0.7521. The average genetic similarity coefficients of the 9 accessions from China were higher than the average value of all the germplasms (0.7521), which revealed the narrow genetic basis of these newly naturalized germplasm. Notably, two accessions from Brazil (PI309958) and from Uruguay (PI193144) were the most recent with a maximum genetic similarity coefficient of 0.9804, while two accessions from South Africa (PI409137) and China (SBC006) were the lowest, which was 0.5609.

**Table 1.** The number and level of polymorphism obtained by 47 SRAP primer combinations on 80 prairie grass accessions.

| Primer Group | TNB | NPB | PBB | DP | H | PIC | RP | MI |
|---|---|---|---|---|---|---|---|---|
| Me2 + em3 | 7 | 6 | 85.7% | 0.607 | 0.468 | 0.425 | 4.325 | 2.550 |
| Me3 + em2 | 6 | 3 | 50.0% | 0.913 | 0.417 | 0.210 | 0.775 | 0.630 |
| Me4 + em1 | 10 | 7 | 70.0% | 0.766 | 0.499 | 0.337 | 3.175 | 2.359 |
| Me4 + em8 | 11 | 11 | 100.0% | 0.854 | 0.472 | 0.350 | 5.350 | 3.850 |
| Me4 + em16 | 11 | 9 | 81.8% | 0.779 | 0.498 | 0.376 | 4.925 | 3.384 |
| Me4 + em20 | 9 | 9 | 100.0% | 0.952 | 0.343 | 0.229 | 2.750 | 2.061 |
| Me5 + em1 | 9 | 8 | 88.9% | 0.772 | 0.499 | 0.328 | 3.750 | 2.624 |
| Me6 + em6 | 12 | 8 | 66.7% | 0.891 | 0.443 | 0.394 | 5.100 | 3.152 |
| Me7 + em7 | 10 | 10 | 100.0% | 0.867 | 0.464 | 0.349 | 5.450 | 3.490 |
| Me7 + em15 | 6 | 4 | 66.7% | 0.532 | 0.432 | 0.218 | 1.025 | 0.872 |
| Me7 + em19 | 10 | 8 | 80.0% | 0.772 | 0.499 | 0.262 | 2.700 | 2.096 |
| Me8 + em3 | 3 | 3 | 100.0% | 0.451 | 0.383 | 0.348 | 1.550 | 1.044 |
| Me8 + em7 | 9 | 7 | 77.8% | 0.655 | 0.485 | 0.344 | 3.575 | 2.408 |
| Me8 + em16 | 5 | 2 | 40.0% | 0.202 | 0.190 | 0.184 | 0.425 | 0.368 |
| Me9 + em1 | 13 | 7 | 53.8% | 0.766 | 0.499 | 0.227 | 1.925 | 1.589 |
| Me9 + em20 | 14 | 13 | 92.9% | 0.629 | 0.476 | 0.331 | 5.850 | 4.303 |
| Me10 + em9 | 15 | 11 | 73.3% | 0.866 | 0.465 | 0.281 | 4.375 | 3.091 |
| Me10 + em16 | 8 | 6 | 75.0% | 0.528 | 0.430 | 0.399 | 3.450 | 2.394 |
| Me11 + em3 | 11 | 10 | 90.9% | 0.812 | 0.491 | 0.354 | 5.375 | 3.540 |
| Me11 + em6 | 13 | 10 | 76.9% | 0.613 | 0.470 | 0.355 | 5.500 | 3.550 |
| Me11 + em8 | 10 | 7 | 70.0% | 0.818 | 0.489 | 0.276 | 2.575 | 1.932 |
| Me12 + em10 | 7 | 6 | 85.7% | 0.775 | 0.490 | 0.311 | 2.550 | 1.866 |
| Me12 + em17 | 10 | 7 | 70.0% | 0.907 | 0.424 | 0.274 | 2.825 | 1.918 |
| Me13 + em9 | 7 | 6 | 85.7% | 0.505 | 0.417 | 0.367 | 3.400 | 2.202 |
| Me13 + em18 | 13 | 8 | 61.5% | 0.758 | 0.500 | 0.342 | 3.725 | 2.736 |
| Me13 + em19 | 12 | 8 | 66.7% | 0.59 | 0.460 | 0.299 | 3.400 | 2.392 |
| Me14 + em3 | 10 | 8 | 80.0% | 0.677 | 0.491 | 0.328 | 3.900 | 2.624 |
| Me14 + em12 | 12 | 10 | 83.3% | 0.777 | 0.498 | 0.343 | 5.200 | 3.430 |
| Me15 + em6 | 15 | 8 | 53.3% | 0.848 | 0.476 | 0.359 | 4.550 | 2.872 |
| Me15 + em13 | 6 | 5 | 83.3% | 0.751 | 0.500 | 0.287 | 2.050 | 1.435 |
| Me16 + em6 | 8 | 4 | 50.0% | 0.784 | 0.498 | 0.362 | 2.175 | 1.448 |
| Me17 + em7 | 9 | 7 | 77.8% | 0.655 | 0.485 | 0.319 | 3.225 | 2.233 |
| Me17 + em13 | 9 | 8 | 88.9% | 0.766 | 0.500 | 0.249 | 2.450 | 1.992 |
| Me17 + em16 | 10 | 5 | 50.0% | 0.468 | 0.394 | 0.347 | 2.700 | 1.735 |
| Me18 + em1 | 8 | 4 | 50.0% | 0.913 | 0.417 | 0.331 | 2.075 | 1.324 |
| Me18 + em2 | 9 | 6 | 66.7% | 0.703 | 0.496 | 0.321 | 3.050 | 1.926 |
| Me18 + em4 | 9 | 5 | 55.6% | 0.782 | 0.498 | 0.383 | 2.975 | 1.915 |
| Me18 + em8 | 10 | 7 | 70.0% | 0.768 | 0.499 | 0.336 | 3.450 | 2.352 |
| Me18 + em10 | 11 | 9 | 81.8% | 0.68 | 0.491 | 0.327 | 4.575 | 2.943 |
| Me18 + em17 | 7 | 6 | 85.7% | 0.684 | 0.492 | 0.307 | 2.350 | 1.842 |
| Me18 + em18 | 13 | 11 | 84.6% | 0.782 | 0.498 | 0.363 | 5.825 | 3.993 |
| Me18 + em19 | 9 | 8 | 88.9% | 0.574 | 0.453 | 0.288 | 3.250 | 2.304 |
| Me19 + em1 | 11 | 9 | 81.8% | 0.754 | 0.500 | 0.265 | 3.075 | 2.385 |
| Me19 + em14 | 14 | 11 | 78.6% | 0.695 | 0.495 | 0.303 | 4.300 | 3.333 |
| Me20 + em6 | 11 | 9 | 81.8% | 0.749 | 0.500 | 0.300 | 3.675 | 2.700 |
| Me20 + em10 | 12 | 9 | 75.0% | 0.850 | 0.475 | 0.279 | 3.625 | 2.511 |
| Me20 + em20 | 6 | 2 | 33.3% | 0.836 | 0.482 | 0.324 | 0.875 | 0.648 |
| sum | 460 | 345 | | | | | | |
| mean | 9.8 | 7.3 | 75% | 0.753 | 0.465 | 0.317 | 3.387 | 2.314 |

TNB: total of bands; NPB: the number of polymorphic bands; PBB: percentage of polymorphic bands; PIC: polymorphic information content; MI: marker index; RP: resolving power; H, heterozygosity; DP: discriminating power.

**Table 2.** Genetic diversity estimation for six geographical groups of prairie grass accessions.

| Geographical Origin | N | Na | Ne | I | He | uHe |
|---|---|---|---|---|---|---|
| South America | 22 | 1.936 | 1.556 | 0.484 | 0.323 | 0.330 |
| North America | 14 | 1.928 | 1.585 | 0.498 | 0.335 | 0.347 |
| Africa | 13 | 1.887 | 1.558 | 0.476 | 0.319 | 0.332 |
| Europe | 8 | 1.594 | 1.476 | 0.403 | 0.273 | 0.309 |
| Oceania | 8 | 1.728 | 1.507 | 0.431 | 0.289 | 0.291 |
| Asia | 15 | 1.472 | 1.325 | 0.295 | 0.194 | 0.200 |
| Mean | 13.333 | 1.757 | 1.501 | 0.431 | 0.289 | 0.302 |

N: accessions number; Na: the allele number; Ne: effective number of alleles; I: Shannon information index; He: Expected heterozygosity; uHe: Unbiased expected heterozygosity.

### 3.2. *Hierarchal Clustering, Pcoa, Neighbor-Net and Population Structure Analysis*

Hierarchal clustering of 80 accessions based on their DICE similarity coefficients was performed using UPGMA clustering. The dendrogram indicated that all of the prairie grass accessions could be divided into five main clusters (Pop1, 2, 3, 4 and 5) (Figure 1). Overall, the UPGMA clustering patterns have little correlation with geographical origin. Pop1 consisted of 20 accessions from six countries including South Africa, Argentina, Brazil, France, the United States and China. Pop2 contained 36 accessions, including all three Uruguayan samples, all three Indian germplasm, all two Japanese samples and the rest from other ten countries. Accessions from Asia are mostly gathered in this group. Pop3 had 8 germplasms, of which four are from North America. Pop4 consisted of 11 accessions, most of which were from South America. Pop5 contained 5 accessions from four countries including The United States, South Africa, Ethiopia and France. Based on the Dice similarities matrix, PCoA was executed and the scatter plot was generated to visualize the dispersion of the accessions based on the main differentiation factors (Figure S3). The first three coordinate axes (eigenvectors) captured 41.75% of the total molecular variation, which accounted for 22.75%, 13.58%, and 5.42% of the observed variation, respectively. Based on accessions position relative to the two coordinate axes, a two-dimensional plot revealed a nearly consistent grouping of entire germplasm collection with the UPGMA dendrogram. Based on the data generated from the morphological traits, the dendrogram was constructed and divided the studied accessions into three major clusters at an average distance of 0.81, showing an independent grouping irrespective of their improvement status and geographical distribution or country (Figure S4).

Next, we analyzed the correlations between phenotypic traits and Q values with Mantel test. Nine phenotypic traits were all significantly correlated with the Q matrix, and the stem diameter (SD) has the greatest correlation with Q matrix ($r = 0.41$, $p < 0.0001$). Results of the Mantel test and Procrustes analysis (Figure S5) showed a moderate positive correlation between the distance matrices based on the morphological and molecular data sets (Mantel: $r = 0.524$, $p < 0.001$; Procrustes: $M^2 = 0.2178$, $p < 0.001$). Thus, a clustering pattern of prairie grass accessions based on morphological traits was associated with that derived from the SRAP data.

In order to further understand the relationship between the studied accessions, the STRUCTURE software was used to analyze the marker information by a Bayesian-based model. According to Evanno's method, the optimal K value was 3 (Figures 1 and S6), showing the most suitable number of subgroups to be three, namely, three genetic memberships (Table S4). Supposing that accessions with a membership coefficient (Q value) of 0.8 or more were considered as pure, 86.25% of the studied germplasm were considered to corresponding pure subgroups while the rest 13.75% were categorized as admixed subpopulation. Of which, European accessions are of pure genetic backgrounds, germplasm from North America had the largest proportion of mixed sources (Table S5). The Q values of five UPGMA Pops were also analyzed (Table S6). The results showed that Pop1, Pop4 and Pop5 germplasm were assigned to different pure origins (color red and blue). All eight accessions of Pop3 and two from Pop2 were considered to have admixed membership.

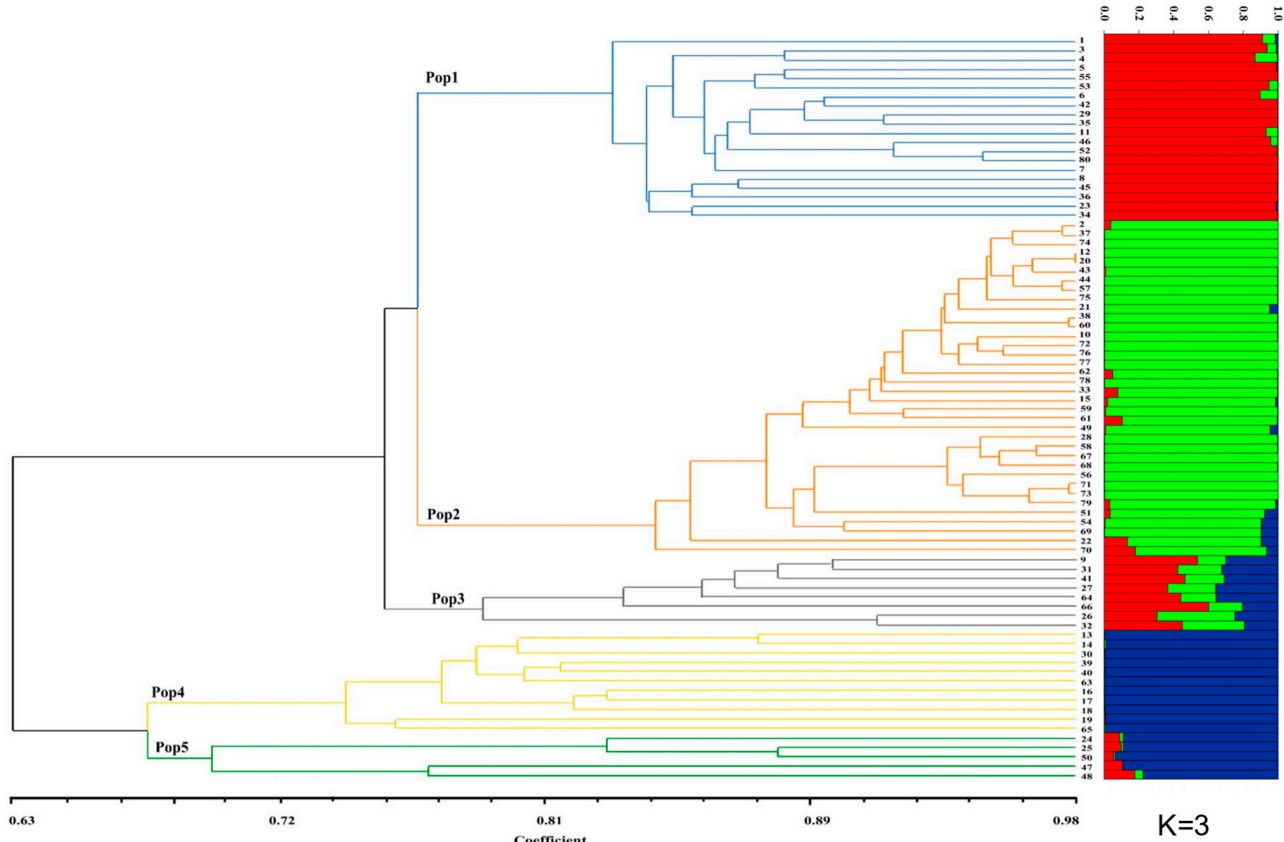

**Figure 1.** The UPGMA tree of prairie grass accessions and genetic background using a Bayesian STRUCTURE analysis at K = 3 (membership = 3).

The Neighbor-Net of the 80 prairie grass accessions (Figure S7, implemented in SplitsTree4) revealed essentially the same separation patterns of 5 deduced groups from the UPGMA tree, PCoA and STRUCTURE analysis. It should be noted here that the split Network diagram contains some small "boxes" which indicates high resolution and complexity of phylogenetic relationships among accessions and necessitates further analysis.

### 3.3. Genetic Structure of Inferred Clustering Groups with UPGMA and the Geographic Groups

All prairie grass accessions were divided into six geo-groups based on geographical origin. North America group had the highest gene diversity indices (He = 0.335) followed by the South American and African groups (He = 0.323 and 0.319, respectively), while the Asian groups had the lowest (He = 0.194) (Table 2). A similar observation was recorded for Shannon index (I) and South American and African groups had the highest (0.498 and 0.484, respectively), while Asian groups had the lowest (0.295) values for this parameter. The AMOVA analysis showed that generality variance occurring within geo-groups contributed 95% of the total variation, and only 5% of total variation was caused by differences among geo-groups with a high significance ($p < 0.001$) (Table 3). The overall Fst (fixation index) values among geo-groups was 0.045, showing a very low genetic differentiation among geo-groups of accessions. The pairwise Fst provided an evaluation of genetic distances between the six geo-groups. The highest differentiation (Fst = 0.172) was observed between geo-group North America and Asia, followed by that between South American and Asia (Table S7).

**Table 3.** Analysis of molecular variance (AMOVA) based on SRAP markers for geographical groups of prairie grass accessions.

| Source of Variation | df | SS | MS | Est. Var. | Fst | PMV (%) |
|---|---|---|---|---|---|---|
| Among geo-groups | 5 | 434.307 | 86.861 | 2.548 | 0.045 | 5% |
| Within geo-groups | 74 | 3977.906 | 53.755 | 53.755 | | 95% |
| Total | 79 | 4412.213 | | 56.303 | | 100% |

df: degree of freedom; SS: square deviation; MS: mean square deviation; Est.Var: exist variance; Fst: coefficient of genetic differentiation; PMV: Percentages of molecular variance.

In addition, diversity components of five inferred Pops identified in UPGMA tree and PCoA were estimated by AMOVA. Among these groups, Pop4 had the highest He value and Pop2 and Pop3 had the lowest He value (Table S8). Of the total genetic variance covered by the five Pops, 36% (Table S9) was caused by the variation among Pops while 64% of the mutation was caused by variation among accessions within Pops. These variation components were significantly different from zero ($p < 0.0001$) based on the arrangement test. The average fixation index (Fst) among the five Pops showed high genetic differentiation (Fst = 0.356). The highest differentiation (0.525) was found between Pop2 and Pop4 and the lowest (0.042) was found between Pop1 and Pop2 (Table S10).

## 4. Discussion

### 4.1. Genetic Polymorphisms and Discriminating Capacity of the SRAP Primers

The analysis of genetic diversity of plant germplasm is an effective means to explore superior breeding resources and improve breeding efficiency [33]. SRAP, an efficient and simple marker technique, has been proved as more informative than other dominant DNA marker types for genetic diversity detection in many plant species [34–36]. To our knowledge, this is the first study applying SRAP markers to describe the genetic structure and diversity of 80 prairie grass accessions from a wide range of geographical regions. In the current investigation, a relatively high percentage of polymorphic bands (75%) were detected using SRAP primers, which was higher than the proportion of polymorphism reported from some grass species such as *Avena macrostachya* (69.3%) [37] and *Pennisetum purpureum* (72.8%) [38]. This showed that these SRAP markers have a high potential for elucidating the genetic diversity of prairie grass. The mean Nei's gene diversity index (He = 0.289) was higher than the average of monocotyledon from a meta-analysis (0.190) [39], which indicated that the high level of genetic diversity exists in the studied germplasm collection of prairie grass. However, the polymorphism level available are actually heterogeneous with respect to the number of accessions examined, range of geographic origin and improvement status.

In general, the PIC values are suggestive of polymorphic nature at single loci or sum of multiple loci analyzed [40]. The PIC value for SRAP markers in the current study is in the range of 0.184–0.425, with the average of 0.317 (>0.25), also revealed the varied genetic diversity among the global prairie grass accessions. The primary reason for this high diversity is likely due to the self-pollinating properties and broad geographical origins of studied prairie grass accessions. The former led to higher inter-accession (population) and lower intra-accession variability, the latter limited gene flow among distant accessions and further aggravated their genetic heterogeneity [39]. Some primer efficiency indices, including RP and MI, represent the overall utility of a particular primer for characterization and discrimination of large number of accessions. Specifically, the higher their values, the more efficient and informative the primers will be. In current investigation, the average MI (1.348) and RP (1.897) values of the primers used were higher compared to those observed in our previous research by EST-SSR markers (MI = 0.67, RP = 1.14) [12]. The highest MI and RP were obtained by primer pair Me9-Em20 (MI = 5.85, RP = 4.30) and Me18-Em18 (MI = 5.82, RP = 3.99), showing these primers are highly efficient and useful for genetic discrimination for prairie grass germplasm.

*4.2. Genetic Relatedness and Population Structure*

The results of three distance-based clustering analyses performed (UPGMA clustering, SplitsTree and PCoA analyses) are surprisingly consistent, dividing studied accessions into five groups/clusters which appear to have some correlation with their geographical origin. Evidence from previous studies found that the rainfall in the native distribution area could be an important factor contributing to the genetic differentiation of wild prairie grass populations. Namely, a significant positive association was found between humidity/rainfall level and phenotypic diversity in Argentinean populations of prairie grass [13]. The genetic diversity level of distance-based cluster Pop2 and Pop3, most of which were collected from countries with humid climates, was lower than Pop4 containing a considerable number of accessions from countries with drier climates. Nevertheless, it is hard to make a definitive conclusion regarding this issue due to the lack of detailed geographical coordinates and climatic data of the studied accessions. AMOVA and three distance-based analyses demonstrated low geographical sub-structuring among the 80 accessions of prairie grass, since the overall global Fst value (0.045) indicated a very low degree of genetic differentiation between six geo-groups. Nevertheless, diversity indices of South America and North America groups were obviously higher than those of other regions. Considering that areas of greater genetic diversity are usually related to the center of origin of a species, it is not difficult to understand that South America as the origin center of *Bromus catharticus* showed higher germplasm diversity [41] while North America may be the secondary center of origin whose diversity is even slightly higher than that of South America. Moreover, results from the Mantel test and Procrustes analysis detected a significant moderate correlation between Euclidean distance from morphological traits and genetic distance from SRAP markers (Mantel: r = 0.53, $p < 0.0001$; Procrustes: $M^2$ = 0.2178, $p < 0.001$).

In spite of small differences, Bayesian STRUCTURE analysis was basically consistent with the result of three distance-based methods. In the current study, the model-based STRUCTURE analysis divided the *B. catharticus* accessions into three ancestral memberships (K = 3). 86.25% of accessions showed pure ancestry in the STRUCTURE analysis, while mixed ancestry was discovered among few accessions (13.75%) with a proportionate membership value (Q) < 80%, of which 90% (9) were wild germplasm. Ancestral admixture was considered to be associated with exchange of plant germplasm between areas and/or hybridization [20]. The low proportion of accessions with mixed ancestry may be due to the fact that prairie grass is a strong self-pollinating (autogamous) plant with predominantly homozygous off spring produced by selfing. According to Figure 1, except that the distance-based Pop3 corresponds with most of the accessions with admixed ancestry, few mixed accessions are scattered across Pop2 and Pop5. Hence, Bayesian model-based analysis could assign each accession to a hypothetical ancestral group(s) without any a priori information as well as disclose the admixture that were non-obvious through distance-based clustering methods [20,42]. The proportion of pure origin of South America's germplasm was higher than that of Asia, which is consistent with the results of our previous works on prairie grass [12].

**5. Conclusions**

In the present study, we evaluated the genetic diversity and molecular characterization of worldwide prairie grass accessions using SRAP markers. Cluster analysis based on DICE similarity coefficients grouped all 80 prairie grass accessions into five main clusters. The results of PCoA, SplitsTree and UPGMA cluster analysis were consistent with each other. The AMOVA suggested that most of the generality variance existed within geo-groups, and the Fst among geo-groups was 5%. The genetic diversity indices of the accessions from South America and North America were the highest, which could be considered as the excellent germplasms and should be further protected and utilized. In summary, our study provided a comprehensive insight of the diversity in prairie grass from the perspective of SRAP markers, which could further promote its utilization and breeding process.

**Supplementary Materials:** The following are available online at https://www.mdpi.com/article/10.3390/agronomy11102054/s1, Table S1: Information of 80 prairie grass accessions used in this study. Table S2: Nine morphological traits measured in 80 prairie grass. Table S3: Information of SRAP primers. Table S4: Q values obtained at K = 3. Table S5: Distribution of the Q value of prairie grass germplasm in the 6 geographical groups. Table S6: Distribution of the Q value of prairie grass germplasm in the 5 inferred clusters by UPGMA and Neighbor-Net tree. Table S7: Pairwise Population PhiPT Values among six geographical groups of prairie grass accessions. Table S8: Genetic diversity estimates for five inferred Pops of prairie grass accessions. Table S9. Analysis of molecular variance (AMOVA) based on SRAP markers for five inferred Pops of prairie grass accessions. Table S10: Pairwise Population PhiPT Values among five Pops of prairie grass accessions. Figure S1: Global distribution of the 80 prairie grass accessions used in this study. Figure S2: Amplification profiles of 80 prairie grass accessions with primer combination Me4 and Em8, ME7+EM15, ME14+EM12 and ME17+EM13 (accessions 1 to 80 from left to right). Figure S3: Two-dimensional plot based on principal coordinate analysis of 80 prairie grass accessions based on the SRAP markers. Figure S4: UPGMA dendrogram and heat map delineating 80 prairie grass accessions based on nine morphological traits. Figure S5: Procrustes analysis of the correlation between genotype and phenotypic traits ($M^2$ = 0.2178 $p$ < 0.01). Figure S6. Population structure analysis of 80 prairie grass accessions (A) Estimates of the rate of the slope of the log probability curve (DK) plotted against K (B) Plot of the Ln probability of data, LnP(D), averaged over the replicates. Figure S7. Neighbor-Net graph of 80 prairie grass accessions based on SRAP markers computed with the SplitsTree4 program.

**Author Contributions:** Conceptualization, Y.L. and J.Y.; Data curation, L.Y. and Z.D.; Formal analysis, L.Y., Z.D. and Y.X. (Yanli Xiong); Funding acquisition, X.M.; Investigation, Y.X. (Yi Xiong); Methodology, Y.X. (Yanli Xiong), J.Y. and J.Z.; Project administration, Y.L.; Resources, Y.X. (Yi Xiong), J.Y. and W.G.; Supervision, J.Z.; Validation, J.Z.; Visualization, J.Z. and Y.L.; Writing—original draft, L.Y. and Z.D.; Writing—review and editing, W.G. and X.M. All authors have read and agreed to the published version of the manuscript.

**Funding:** This research was supported by the earmarked fund for Modern Agro-industry Technology Research System (No. CARS-34) and National Natural Science Foundation of China (3177131276), and Sichuan Beef Cattle Industrial System Innovation Teams of National Modern Agricultural Industrial Technology System, SCCXTD-2020-13.

**Institutional Review Board Statement:** Not applicable.

**Informed Consent Statement:** Not applicable.

**Data Availability Statement:** Data is contained within the article or Supplementary Materials.

**Acknowledgments:** We thank the laboratory staff in the Department of Grassland Science and Technology College, Sichuan Agricultural University.

**Conflicts of Interest:** The authors declare no conflict of interest.

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
