# Peer review of "Genetic Diversity and Molecular Characterization of Worldwide Prairie Grass (Bromus catharticus Vahl) Accessions Using SRAP Markers"

_agronomy, doi:10.3390/agronomy11102054_

Round 1
Reviewer 1 Report
The authors describe the application of a well-established protocol (SRAP) for the analyses of genetic diversity in prairie grass (Bromus carthaticus).
The paper is generally well written, though further attention is needed to the English grammar, italicization of scientific names and details pertaining to the STRUCTURE analysis which is incongruent with other results.
Re STRUCTUE analyses: The number of independent replicates for each K value is quite low (3), I recommend a greater value (~20) which is in line with Evanno et al. (2005) and more recent reviews of the reliability of structure based approaches. The LOCPRIOR option may improve group assignment. If the authors did follow Evanno’s method as suggested in the discussion, it should be mentioned in the materials and methods. The authors should present supporting evidence such as a scree plot, delta K plot and the mean L(K) +/- standard deviation generated from STRUCTURE HARVESTOR within the supplemental materials to confirm the results with the reader. Contradictory statements are made on how well the STRUCTURE analyses correspond with the other analyses. See page 6, where STRUCTURE results are mentioned as being in line with UPGMA/PCoA results versus (5 clusters) when three ancestral groups are presented.
How many PCoA axes were generated? Do additional axes correspond with the other population structure metrics?
The authors describe relatively high polymorphism within Bromus carthaticus, as this species is hexaploid are the authors confident in these results? I would like to see the polyploidy nature of the species addressed and the associated implications on estimates of genetic diversity.
The Mantel test showed a relatively strong correlation between population structure and the phenotypic distance matrix. Are there specific correlations present between phenotypic traits and Q values? While the authors were unable to examine the environmental-population structure correlations, analyzing the phenotype-population structure correlations would be useful and tell a more complete story about the germplasm. Figure S1 suggests phenotypic divergence between three groups. How well do the results of the structure analysis correspond with the phenotypic distances? Is there a phenotype that corresponds with these patterns?
Figure and table titles/captions need to be more descriptive throughout the manuscript. As an example Figure S2 is not informative in it’s current form. Please review the caption or provide a legend to define the clusters so that the message is clear to the reader.
Please clarify the following:
In conclusion, prairie grass was outcrossing species and cross-pollination may lead to admixture of alleles from neighboring regions. The low proportion of accessions with mixed ancestry may be due to the fact that prairie grass is a strong self-pollinating (autogamous) plant with predominantly homozygous off spring produced by selfing.
Author Response
All the modified parts have been marked in the manuscript with yellow backgrounds. The responses to the reviewer's questions are as follows.
Comment 1) The authors describe the application of a well-established protocol (SRAP) for the analyses of genetic diversity in prairie grass (Bromus carthaticus). The paper is generally well written, though further attention is needed to the English grammar, italicization of scientific names and details pertaining to the STRUCTURE analysis which is incongruent with other results.
Response: Thank you for your comment. We have revised the English grammar and italicization of scientific names question.
Comment 2) Re STRUCTUE analyses: The number of independent replicates for each K value is quite low (3), I recommend a greater value (~20) which is in line with Evanno et al. (2005) and more recent reviews of the reliability of structure based approaches. The LOCPRIOR option may improve group assignment. If the authors did follow Evanno’s method as suggested in the discussion, it should be mentioned in the materials and methods. The authors should present supporting evidence such as a scree plot, delta K plot and the mean L(K) +/- standard deviation generated from STRUCTURE HARVESTOR within the supplemental materials to confirm the results with the reader.
Response: Thanks for the reviewer’s precious suggestion. Indeed, three independent replicates for each K value is quite low, thus we re-analyze the genetic structure of the tested accessions with each run values of 20. The results also showed that the 80 accessions possessed three genetic membership (K=3) though slight difference of Q values was observed between run values of 3 and 20. The relevant parts of the materials and methods have been modified and the supporting file has been updated in the appendix.
Comment 3) Contradictory statements are made on how well the STRUCTURE analyses correspond with the other analyses. See page 6, where STRUCTURE results are mentioned as being in line with UPGMA/PCoA results versus (5 clusters) when three ancestral groups are presented.
Response: Thank you for your comment. According to the Evanno's method, the optimal K value was 3, which means the tested 80 accessions possessed three genetic memberships.. However, other analysis (UPGMA/PCoA) results suggested all of prairie grass accessions could be divided into five main clusters. It seems that the results of Structure and UPGMA/PCoA analysis were contradictory. However, accessions possessing the admixed membership in the STRUCTURE analysis are basically all gathered in pop3, and the genetic backgrounds of pop1, pop2, and pop4 (color red, green and blue) are all single and different. Furthermore, considering the algorithm differences of those three analysis, it’s reasonable to obtain the different results.
Comment 4) How many PCoA axes were generated? Do additional axes correspond with the other population structure metrics?
Response: Thank you for your comment. A total of seventy-nine PCoA axes were generated. Because the top three axes could totally explain the variations (41.75%) sufficiently, we only used those components to reveal the population structure of our accessions.
Comment 5) The authors describe relatively high polymorphism within Bromus carthaticus, as this species is hexaploid are the authors confident in these results? I would like to see the polyploidy nature of the species addressed and the associated implications on estimates of genetic diversity.
Response: Thank you for pointing out this question and then we checked the relevant literature. According to Ding et al's analysis of genetic diversity of haploid and diploid Asparagus schoberioides Kunth by AFLP, the genetic diversity of diploid population was higher than that of haploid population [1]. Wan et al used SSR to analyze the genetic diversity of diploid and tetraploid Dactylis glomerata and found that tetraploid group had higher genetic diversity than diploid group [2]. Zhong et al used EST-SSR and IT-ISJ to study the genetic diversity of tetraploid and octaploid Panicum virgatum, the two molecular markers concluded that the genetic parameters of tetraploid were higher than those of octaploid [3]. In conclusion, the ploidy of different species has different implications on genetic diversity, and there is no study on this part of Bromus catharticus.
References:
[1] Ding, H.Y.; Sui, Z.H.; Zhong, J.; Zhou, W.; Wang, Z.X. Analysis and Comparison on Genetic Diversity of Haploid and Diploid Gracilaria lemaneiformis Polulations from different Places of Qingdao by AFLP. Journal of Ocean University of China 2012, 42(1-2):099-105
[2] Wang, G. Genetic diversity of diploid and tetraploid Dactylis glomerata and their F1 hybrids identification. Sichuan Agricultural University, Chengdu, Sichuan, 2011
[3] Zhong, M.; Zhou, S.F.; Zhang, X.Q.; Huang, X.; Yan, H.D.; Huang, L.K. Genetic diversity assessment of tetraploid and octaploid switch grasses using EST-SSR and IT-ISJ molecular markers. Acta Prataculturae Sinica 2016, 25(10):113-123
Comment 6) The Mantel test showed a relatively strong correlation between population structure and the phenotypic distance matrix. Are there specific correlations present between phenotypic traits and Q values? While the authors were unable to examine the environmental-population structure correlations, analyzing the phenotype-population structure correlations would be useful and tell a more complete story about the germplasm. Figure S1 suggests phenotypic divergence between three groups. How well do the results of the structure analysis correspond with the phenotypic distances? Is there a phenotype that corresponds with these patterns?
Response: Thank you for pointing out this question. We analyzed the phenotypic traits and Q values correlations with Mantel test, nine phenotypic traits were significantly correlated with Q matrix, and the stem diameter (SD) has the greatest correlation with Q matrix (r = 0.41, P < 0.0001).
Comment 7) Figure and table titles/captions need to be more descriptive throughout the manuscript. As an example Figure S2 is not informative in it’s current form. Please review the caption or provide a legend to define the clusters so that the message is clear to the reader.
Response: Thank you for your suggestion. we have modified the relevant information in Figure and table titles/captions.
Comment 8) Please clarify the following:
In conclusion, prairie grass was outcrossing species and cross-pollination may lead to admixture of alleles from neighboring regions. The low proportion of accessions with mixed ancestry may be due to the fact that prairie grass is a strong self-pollinating (autogamous) plant with predominantly homozygous off spring produced by selfing.
Response: Thank you for pointing out this question, indeed, the following is our revised content: The low proportion of accessions with mixed ancestry may be due to the fact that prairie grass is a strong self-pollinating (autogamous) plant with predominantly homozygous off spring produced by selfing.
Reviewer 2 Report
Overall the results of the study can help in the future breeding and cultivar development efforts in Prairie grass (Bromus catharticus Vahl). Most of the comments have been provided in the attached word format of the manuscript. Abstract can be improved. Please also include a SRAP gel image in the manuscript. The conclusion section needs rewriting.

Author Response
All the modified parts have been marked in the manuscript with yellow backgrounds. The responses to the reviewer's questions are as follows.
Comment 1) Page 1: line 3: add 'are' before 'distributed'
Response: Thank you for your comment. I have revised in the manuscript in line 3.
Comment 2) Page 2: line 4 not 'researches' change it to "research''
Response: Thank you for your comment. I have revised in the manuscript in line 4.
Comment 3) Page 2: line 18: Not 'lake' but change it to 'lake'
Response: Thank you for your comment. I have revised in the manuscript in line 18.
Comment 4) Page 2': '(25/15°C day/night' correct space issues
Response: Thank you for your comment. I have revised in the manuscript.
Reviewer 3 Report
Page 1: line 3: add 'are' before 'distributed'
Page 2: line 4 not 'researches' change it to "research''
Page 2: line 18: Not 'lake' but change it to 'lake'
Page 2': '(25/15°Cday/night' correct space issues
Please get minor editing errors and English grammar issues corrected.
Author Response

(The authors gave the same response as above.)

Reviewer 4 Report
Dear Authors,
Good job. Manuscript written very well. I only have a few minor comments.
In the description of the material it should be explained what is the difference between cultivated accessions and cultivars. Can cultivated forms be considered as landraces? To illustrate the diverse origins of accessions, it would be useful to have a contour map with infill indicating the number of accessions originating from a given region. There is no description of how morphological traits were evaluated and what those traits were. Formulas in data analysis should be written using the formula wizard. They will be more readable. Please add an ad hoc graph of measure ΔK to the results section and it could be included in a supplement. Have you checked whether the PCoA clustering/dendrogram shows any relationship with any of the traits studied? With such a significant correlation you should be able to see some relationship. Generalized Procrustes analysis (GPA) may also be of value to obtain a consensus configuration for genotype and phenotypic traits and describe variation more comprehensively. The figures in the supplement could be larger.
Best regards,
Rev.
Author Response
All the modified parts have been marked in the manuscript with yellow backgrounds. The responses to the reviewer's questions are as follows.
Comment 1) In the description of the material it should be explained what is the difference between cultivated accessions and cultivars. Can cultivated forms be considered as landraces?
Response: Thank you for your comment. The cultivated accessions here refer to the variety, which has been explained in the article. Thus, cultivated forms can’ t be considered as landraces ..
Comment 2) To illustrate the diverse origins of accessions, it would be useful to have a contour map with infill indicating the number of accessions originating from a given region.
Response: Thank you for your suggestion. we have added the contour map to the appendix.
Comment 3) There is no description of how morphological traits were evaluated and what those traits were.
Response: Thank you for your comment, our method is as follows. Related content has also been added to materials and methods and appendix. Field data were recorded in the experimental field of Sichuan Agricultural University (Yaan Province, China) during the prairie grass growing season (from May to September 2014). Nine morphological traits were scored. The measured traits were plant height (PH, cm), length of first internode (LFI, cm), length of flag leaf (LFL, cm), length of second upper leaf (LSUL, cm), stem diameter (SD, mm), width of flag leaf (WFL, cm), width of second up per leaf (WSUL, cm), dry matter yield (DMY, g plant-1) and tiller number (TN). Among them, PH, SD, LFI, LFL, LSUL, WFL, WSUL were measured 3 times per plant. The DMY and TN per plant can only be measured once.
Comment 4) Formulas in data analysis should be written using the formula wizard. They will be more readable.
Response: Thank you for pointing out this question, we have used the formula wizard in the article.
Comment 5) Please add an ad hoc graph of measure ΔK to the results section and it could be included in a supplement.
Response: Thank you for your suggestion. we have added the ad hoc graph of measure ΔK to the appendix.
Comment 6) Have you checked whether the PCoA clustering/dendrogram shows any relationship with any of the traits studied? With such a significant correlation you should be able to see some relationship.
Response: Thank you for your comment. The PCoA clustering/dendrogram shows some relationship with the traits studied, such as the morphology of pop1 germplasms were similar awfully (especially for low dry matter yield, tiller number and length of first internode). The number of tillers of accessions in Pop2 were lower than other pops. The two germplasms of Pop4 and three germplasms of Pop5 are resemble in morphology and belong to one class in morphology clustering.
Comment 7) Generalized Procrustes analysis (GPA) may also be of value to obtain a consensus configuration for genotype and phenotypic traits and describe variation more comprehensively. The figures in the supplement could be larger.
Response: Thank you for your suggestion. We have added Generalized Procrustes analysis (GPA) to the manuscript